# Multi-Omics Approaches in Oil Palm Research: A Comprehensive Review of Metabolomics, Proteomics, and Transcriptomics Based on Low-Temperature Stress

**DOI:** 10.3390/ijms25147695

**Published:** 2024-07-13

**Authors:** Jerome Jeyakumar John Martin, Yuqiao Song, Mingming Hou, Lixia Zhou, Xiaoyu Liu, Xinyu Li, Dengqiang Fu, Qihong Li, Hongxing Cao, Rui Li

**Affiliations:** 1National Key Laboratory for Tropical Crop Breeding, Chinese Academy of Tropical Agricultural Sciences, Haikou 571101, China; jeromejeyakumarj@gmail.com (J.J.J.M.); 17638596793@163.com (Y.S.); asd974692216@163.com (M.H.); lxzhou@catas.cn (L.Z.); liuxy86@catas.cn (X.L.); lixinyu@catas.cn (X.L.); fudq@catas.cn (D.F.); liqihong@catas.cn (Q.L.); 2Coconut Research Institute, Chinese Academy of Tropical Agricultural Sciences, Wenchang 571339, China; 3School of Life Sciences, Henan University, Kaifeng 475001, China

**Keywords:** oil palm, low temperature, omics, abiotic stress, stress tolerance

## Abstract

Oil palm (*Elaeis guineensis* Jacq.) is a typical tropical oil crop with a temperature of 26–28 °C, providing approximately 35% of the total world’s vegetable oil. Growth and productivity are significantly affected by low-temperature stress, resulting in inhibited growth and substantial yield losses. To comprehend the intricate molecular mechanisms underlying the response and acclimation of oil palm under low-temperature stress, multi-omics approaches, including metabolomics, proteomics, and transcriptomics, have emerged as powerful tools. This comprehensive review aims to provide an in-depth analysis of recent advancements in multi-omics studies on oil palm under low-temperature stress, including the key findings from omics-based research, highlighting changes in metabolite profiles, protein expression, and gene transcription, as well as including the potential of integrating multi-omics data to reveal novel insights into the molecular networks and regulatory pathways involved in the response to low-temperature stress. This review also emphasizes the challenges and prospects of multi-omics approaches in oil palm research, providing a roadmap for future investigations. Overall, a better understanding of the molecular basis of the response of oil palm to low-temperature stress will facilitate the development of effective breeding and biotechnological strategies to improve the crop’s resilience and productivity in changing climate scenarios.

## 1. Introduction

Due to their immobile nature, plants face various challenges from both living and non-living factors in their environment throughout their life cycle, which can have negative effects on their growth and productivity [1]. Biotic stress, caused by pathogens such as viruses, bacteria, and insects, leads to infections, damage, and reduced crop yields. Pathogens directly affect plant hosts by depleting their nutrients, which can ultimately result in plant death [2]. On the other hand, abiotic stresses, such as salinity, drought, flooding, and extreme temperatures, impair the growth, development, and yield of plants [2]. These factors contribute to a global loss of approximately 50% of major crop plants [3]. To develop strategies that enhance plant defense systems and crop production, it is crucial to have a comprehensive understanding of plant–environment interactions, particularly in terms of biochemistry, which dictates plant responses to environmental stimuli.

The oil palm tree, scientifically known as *Elaeis guineensis* Jacq., is native to tropical Africa and thrives in high humidity and temperature conditions. This tropical tree, often referred to as the “World Oil King”, is a highly competitive source of renewable energy and plays a vital role in oil production [4]. Palm oil production has experienced significant growth over the past two decades, making it the most traded vegetable oil globally. It is anticipated to increase twofold within the forthcoming decade [5]. Temperature plays a crucial role in oil palm production, with optimal yields achieved in regions where the maximum average temperature ranges from 29 to 33 °C and the minimum average temperature ranges from 22 to 24 °C [5]. Low temperatures pose a significant threat to oil palm cultivation, limiting the distribution and large-scale development in many countries and reducing productivity. Low temperatures can negatively affect oil palm growth by causing physical damage or disrupting biochemical and physiological functions, ultimately impacting overall growth and yield. In certain areas, low temperatures significantly limit agricultural productivity, with cold-tolerant species often not yielding the highest results [6]. Temperatures below 20 °C can hinder oil palm growth, while temperatures as low as 12 °C can negatively affect fruit development and oil production in sub-tropical regions [7]. These low-temperature exposures can occur daily or seasonally, with some regions experiencing prolonged periods lasting several months. However, there is limited information available regarding the molecular mechanisms underlying the responses of oil palm plants to low temperature stress.

High-throughput omics technologies have revolutionized data collection from various sources, presenting both challenges and opportunities for the development of customized computational approaches for integrative analysis. These technologies include genomics, epigenomics, transcriptomics, proteomics, and metabolomics [8]. By simultaneously analyzing data from different omics platforms for a given biological sample, researchers can obtain a comprehensive understanding of complex biological interactions. The use of machine learning-based approaches, which leverage biological networks, is crucial for robust biomarker modeling [8,9]. This study aims to provide a comprehensive and systematic review of the current state of knowledge on multi-omics approaches in oil palm research, focusing on the application of metabolomics, proteomics, and transcriptomics in understanding the molecular mechanisms and responses of oil palm to low-temperature stress, and to explore the potential of these omics approaches in developing cold-tolerant oil palm varieties, improving crop resilience, and ensuring sustainable oil palm production in the face of climate change. This information also supports screening and analysis platforms, providing valuable insights and benefits to the oil palm industry.

## 2. Cold Stress Tolerance in Oil Palm: From Physiological Responses to Molecular Mechanisms

Cold stress has a significant impact on the growth and productivity of oil palm trees. When exposed to low temperatures, different varieties of oil palm exhibit a complex response that involves changes in gene expression and physiological alterations (Figure 1). Cold stress affects the growth, fruit development, and oil production of oil palm, leading to yellowing and withering of young leaves and flowers, as well as a slowdown in flower bud differentiation. Enhancing the cold tolerance of oil palm would expand its cultivation area and increase production, particularly in subtropical regions [7]. Cryopreservation methods have been developed for oil palm zygotic embryos and embryogenic calli, indicating the susceptibility of these tissues to cryoinjury [10]. The expression levels of genes associated with cold stress responses, such as CBFs, are linked to the accumulation of osmolytes and antioxidant enzyme activities in oil palm [11]. Treatment with atmospheric cold plasma has minimal impact on the chemical characteristics of palm oil, but it does lead to a decrease in iodine value and an increase in TOTOX value, indicating modifications in oxidative stability [12].

When oil palm is exposed to cold stress, various biochemical and physiological changes occur at the molecular and cellular levels to support plant growth and survival under such conditions [13]. These changes include alterations in the composition and structure of cell plasma, increased concentrations of soluble sugar, sugar alcohols, and other low-molecular-weight nitrogenous compounds, decreased free water content, and the production of antifreeze proteins. These modifications enable oil palm to better withstand the adverse effects of abiotic stress [7]. The molecular mechanism underlying these physiological changes in oil palm under cold stress is still unknown. However, in Arabidopsis and rice, the CBF-mediated signal transduction pathway plays a central role in cold tolerance [14,15]. In Arabidopsis, transgenic plants that constitutively express CBF1, CBF2, and CBF3 exhibit increased cold tolerance, while down-regulation of CBF expression leads to a decrease in freezing tolerance [16].

Lei et al. (2014) [7] revealed that the expression pattern of CBFs in oil palm differs from that of other plant species. This unique expression pattern suggests that oil palm CBFs may possess distinct regulatory mechanisms and functions in response to cold stress. Additionally, the study identified several COR genes that are regulated by CBFs in oil palm. These findings indicate that the CBF-mediated cold response pathway in oil palm is complex and involves multiple downstream targets [7]. Malondialdehyde (MDA) is a biomarker that indicates cell damage and oxidative stress in oil palm leaves exposed to cold conditions. The increase in MDA content suggests cellular damage. According to Li et al. (2019) [11], when oil palm leaves are subjected to cold temperatures (4 °C, 8 °C, and 12 °C), MDA concentrations initially decrease on the first day, suggesting an initial response to lower temperatures, which may involve changes in membrane integrity and lipid peroxidation. However, MDA levels subsequently increase, particularly in young leaves, and by the eighth day, a significant rise in MDA content is observed. Mature leaves exhibited similar patterns under different cold stress conditions. Li et al. (2019) [11] found that proline content in oil palm leaves significantly increases after exposure to different cold treatments, gradually rising throughout the treatment duration compared to control conditions.

## 3. Multiomic and Genome Sequencing Approaches in Oil Palm

Recent advancements in oil palm research have focused on using multi-omics approaches to better understand stress responses and improve agricultural practices. Studies have demonstrated the effectiveness of Multi-Omics Integration (MOI) in uncovering the molecular mechanisms responsible for traits such as drought and salinity resistance in oil palm plants. Integrative omics analyses, including transcriptomics, proteomics, and metabolomics, have been employed to investigate how oil palm plants respond to stress factors like low-temperature stress, revealing key pathways affected by these conditions. Through these approaches, researchers have identified candidate genes, proteins, and metabolites that play a critical role in enhancing stress tolerance and improving productivity [11,17]. These findings offer valuable insights for breeding programs and precision agriculture applications. The integration of genome, proteome, transcriptome, and metabolome data is proving to be essential in understanding oil palm biosynthesis pathways and enhancing oil yield and quality parameters (Figure 2).

### 3.1. Genomics

Economically important traits in fruit trees, like yield and quality, are likely controlled by multiple multi-allelic genes or a large number of genes. Genomics entails the development of molecular markers for genetic diversity analysis and offers new chances to manipulate quantitative trait locus (QTL) through marker-assisted selection to cultivate enhanced cultivars [18]. Genome-wide association studies (GWASs) use the allelic state of unrelated individuals to detect markers associated with target traits, reducing the need for family-based methods like biparental controlled crosses [19]. GWASs are a fine-scale technique that use association mapping to identify individual markers in linkage disequilibrium (LD) related to target variables, therefore eliminating genetic drag effects and accounting for population structure [20]. This method is more effective than QTLs since it has been extensively studied. Incorporating population structure and kinship information might help to address a fundamental issue in GWASs: spurious correlations between markers and phenotypes [21]. Single nucleotide polymorphisms (SNPs) are a prominent genetic marker for assessing individual variations in a specific area of the genome [22].

Recent improvements have enabled high-throughput and high-density genotyping at reduced costs per marker point for genomic analysis [19]. Next-generation sequencing methods, such as genotyping by sequencing, can identify molecular markers for genomic studies like GWASs [23]. GWASs can also be employed to indirectly detect high yields through other secondary traits not associated with fruit quality, if the two traits are highly correlated. Bai et al. [24] found two putative QTLs for leaf area in oil palm, a characteristic that has a strong correlation with oil yield. Thus, breeders might indirectly select trees with high oil output using MAS by recognizing trees with large leaf area, minimizing the arduous and detrimental phenotyping typically required to evaluate oil yield [24].

Further research has used GWASs using genetic markers other than SNPs. For example, Iwata et al. [25] and Cao et al. [26] used simple sequence repeat (SSR) markers to evaluate the quality of fruit in Japanese pear and peach. The utilization of genetic and genomic analysis for identifying DNA regions closely linked to agronomic traits in crops, also known as molecular markers, can aid in enhancing breeding strategies for crop improvement [27]. Molecular markers are utilized for various purposes by exploiting DNA polymorphism. There are different types of markers, including morphological, biochemical, and DNA-based markers. DNA markers are classified as hybridization-based (RFLP) and PCR-based (RAPD, AFLP, SSR, SNP, EST, etc.), with the microsatellite DNA marker being the most widely used due to its simplicity in PCR and high information content from multiple alleles per locus. These markers are developed from genomic DNA libraries or random PCR amplification. Employing molecular markers for the indirect selection of improved crops can expedite the selection process by alleviating time-consuming direct screening methods under field conditions [28].

### 3.2. Oil Palm Genome Sequencing

Consequently, sequencing the oil palm genome is imperative for deciphering its genetic blueprint and enhancing cultivation methods. In 2013, the Malaysian Palm Oil Board (MPOB) made a significant breakthrough in oil palm research by successfully sequencing approximately 1.5 out of 1.8 gigabases (GBs) of the oil palm genome [29]. This achievement was a result of a collaborative effort involving the sequencing of genomic, bacterial artificial chromosome (BAC), and paired-end linker libraries using Roche/454 GS FLX Titanium and Sanger BAC end sequencing platforms. The raw sequencing data consisted of 46.8 billion base pairs, which were assembled using the Newbler assembly tool. This resulted in the generation of 40,360 scaffolds. The assembly and annotation of the oil palm genome provided valuable insights into the genetic makeup of this important crop. Three years later, Jin et al. [30] successfully reported the sequencing of approximately 1.7 GBs of the dura fruit form genome sequence, marking a major breakthrough in oil palm genomics. This was achieved through de novo assembly using a combination of assembly tools, including SOAPdenovo, ABySS, IDBA, Velvet, Oases, Sparse Assembler, Gossamer, and Allpaths-LG. The pisifera genome was used as a reference to facilitate the assembly process, resulting in the generation of 10,971 scaffolds.

The sequencing of the oil palm genome has led to the identification of three crucial genes that play a significant role in determining key traits in oil palm. These genes, SHELL, VIRESCENS, and MANTLED genes, are the three genes that were successfully discovered after the oil palm genome was sequenced. SHELL genes determine the fruit form (pisifera, dura, and tenera) [31]. The fruit color, or VIRESCENS, genes were identified by Singh et al. (2014) [32] and are related to fruit maturation traits that are useful in harvesting. The MANTLED gene is important for the detection of somaclonal variant fruit [33]. Wang et al. [34] have triumphantly presented a reference genome of the oil palm Dura at the chromosome level, exhibiting high quality and encompassing a substantial fraction of the estimated genome size. This achievement offers profound insights into the evolutionary trajectory and genetic advancements possible for palms. Furthermore, this genome sequence has served as a cornerstone for the assembly of preliminary genome sequences of other palm species, including date palm and coconut, at the chromosomal level [34]. The accessibility of these genomic resources holds the potential to expedite genetic improvement initiatives and deepen our comprehension of palm evolution. For instance, these resources can be harnessed to pinpoint candidate genes and genomic locales linked to favorable traits, such as fruit color in oil palm [35], thereby advancing the field of palm genomics.

### 3.3. Importance Development of Oil Palm Genome Sequencing

Genomics holds a crucial position in advancing oil palm breeding programs to satisfy the escalating demand for vegetable oil in a sustainable manner. The sequencing of the oil palm genome has facilitated the identification of genes related to important traits such as fatty acid composition, disease resistance, and nutritional value enhancement. Leveraging various genomic tools and next-generation technologies, researchers aim to enhance oil palm’s yield and quality. Among the key genomic approaches are QTL mapping on chromosomes 4, 10, 12, and 15 [36], as well as the utilization of post-genomic tools such as transcriptomics, proteomics, and metabolomics to evaluate traits and understand fruit ripening and fatty acid synthesis [37].

Genomics-assisted selections and genetic engineering techniques have played a significant role in developing oil palm varieties with novel traits and higher oil productivity [38]. The use of genomic selection and other genomics-guided breeding approaches can accelerate breeding programs and reduce the time and cost of developing new varieties. These approaches can enable the selection of superior genotypes based on genomic data and facilitate the testing and deployment of new varieties with improved traits. Furthermore, genomic selection (GS) has emerged as a transformative improvement in oil palm breeding, augmenting selection intensity and genetic gain through efficient prediction models [39]. Despite these challenges, the prospects for the successful implementation of oil palm genome sequencing in agricultural research and breeding programs are promising. Advances in sequencing and assembly technologies, the development of new genomic resources, and the use of genomics-guided breeding approaches can help to overcome these challenges and enable the development of new oil palm varieties with improved traits.

### 3.4. Challenges in Oil Palm Genome Sequencing

Oil palm possesses a highly intricate genome, characterized by a substantial proportion of repetitive sequences, which significantly complicates its assembly and annotation processes. In the realm of oil palm genome editing, base editing technology encounters formidable obstacles due to the genome’s intricate nature and the challenges posed by tissue culture and genetic transformation methods [40]. Furthermore, the lengthy breeding cycle and the constrained genetic diversity within existing breeding populations pose considerable hurdles for traditional breeding programs [41]. The limitations of conventional breeding, tissue culture, and transgenic research underscore the imperative need for comprehensive genome sequences to address these challenges [42]. Efforts to address these challenges involve utilizing advanced technologies like CRISPR/Cas9 for efficient genome editing to introduce desired traits without losing common ones. Additionally, the application of paired-end ddRAD-sequencing has enabled the discovery of high-quality single nucleotide polymorphisms (SNPs) in diverse oil palm populations, facilitating the recombination of favorable alleles through molecular breeding and selection [4]. The successful completion of the oil palm genome sequencing project continues to face challenges, including the scarcity and fragmentation of existing palm genome sequences as well as the inherent complexities of the palm genome itself. The intricacies of the genome and the low efficiency of genetic transformation pose significant barriers to accurate target gene screening, thereby hampering the precision and efficacy of genome editing [43]. These challenges underscore the need for constant technological and methodological advancements to overcome these limitations. In summary, the challenges surrounding oil palm genome sequencing encompass its complex genome structure, challenges in tissue culture and genetic transformation, limited genetic variations, and the urgent demand for comprehensive genome sequences [44].

### 3.5. Progress in Oil Palm Genome Sequencing

Despite the significant challenges, noteworthy advancements have been achieved in oil palm genome sequencing. In oil palm, Singh et al. [29] and Jin et al. [30] predict 34,082 and 36,105 genes for pisifera and dura oil palm, respectively. The results obtained from Singh et al. [31] used combination prediction outputs from Glimmer [45] and SNAP [46] programs. NCBI ELAEIS GUINEENSIS Annotation Release 101 (https://www.ncbi.nlm.nih.gov/genome/annotation_euk/Elaeis_guineensis/101/) (accessed on 6 January 2017) reported 26,258 genes predicted from Gnomon (the NCBI eukaryotic gene prediction tool). This number of genes is slightly reduced from the Seqping pipeline [47] and FGenesh++ whereby 26,087 of genes were identified and reported by Chan et al. [48]. Some of the scaffolds are contaminants, so we then reduced the predicted genes to 26,059. Gene model prediction via Gnomon (the NCBI eukaryotic gene prediction tool) not only used data from closely related organisms but also included all other studied organisms and well-studied genomes. In this case, this will refine the gene model sets available. In MPOB, Seqping was developed by integrating three different software using ab initio and evidence-based approaches: GlimmerHMM (http://ccb.jhu.edu/software/glimmerhmm/) (accessed on 31 May 2024) [45], SNAP (http://snap.stanford.edu/snap/download.html) (accessed on 31 May 2024), and AUGUSTUS (http://augustus.gobics.de/) (accessed on 31 May 2024) [49]. The outputs were combined using MAKER2 (http://www.yandell-lab.org/software/maker.html) (accessed on 31 May 2024) [50] which uses transcriptome data as evidence for the gene models predicted.

Since then, several enhanced versions of the genome have been made available, offering increasingly accurate and comprehensive assemblies. The most recent iteration, Pisifera_322, released in 2020, stands as a high-quality reference genome for oil palm. Boasting a contig N50 length of 2.8 Mb and a scaffold N50 length of 105.5 Mb, it represents one of the most comprehensive and precise assemblies to date [51].

However, there are promising prospects for addressing these challenges. CRISPR/Cas9-based gene editing techniques show potential for introducing desired traits into the oil palm genome [52,53]. Additionally, the high-quality chromosome-level reference genome of an oil palm Dura has been reported, providing crucial genomic resources for genetic enhancement [54]. The insights gained from this study could potentially accelerate genetic improvement efforts and deepen our understanding of palm evolution [53].

### 3.6. Applications of Oil Palm Genome Sequencing

The oil palm genome sequence holds immense value in oil palm research and breeding efforts. It empowers researchers to achieve the following:(a)Identify genes that are intricately linked to critical agronomic traits, including yield, disease resistance, and stress tolerance.(b)Develop molecular markers, which serve as powerful tools for marker-assisted selection in breeding programs, enhancing the efficiency and precision of cultivar development.(c)Explore the genetic diversity and population structure of oil palm, providing insights into the species’ natural variability and breeding potential.(d)Investigate the evolutionary history and domestication process of oil palm, furthering our understanding of its adaptation and improvement over time.(e)Develop genome-editing tools that enable targeted improvements in oil palm cultivars, potentially enhancing their productivity, resilience, and adaptability.

Oil palm genome sequencing is a pivotal step understanding the genetic foundations of key agronomic traits. By delving into the genome, we can facilitate targeted breeding programs that aim to enhance the productivity of oil palm. This scientific endeavor promises to revolutionize the industry, paving the way for more sustainable and efficient cultivation practices and ultimately contributing to the global demand for palm oil.

## 4. QTLs Contributing to Cold Stress Tolerance

In the oil palm, QTLs that enhance cold stress tolerance have been pinpointed through the development and thorough characterization of Expressed Sequence Tag–Simple Sequence Repeat (EST-SSR) markers. These markers were derived from expressed sequences that exhibited a minimum two-fold change, either up or down, in their expression patterns when exposed to cold stress. Xiao et al. [55] discovered that among the 442 primer pairs situated proximal to SSR repeats that respond to cold stress, 182 polymorphic markers were effectively produced. This remarkable success rate in generating markers with polymorphic variations underscores their potential value in genetic studies pertaining to plants’ cold response. Using in silico mapping techniques, 137 of the 182 polymorphic SSR markers were allocated to specific positions throughout the 16 chromosomes of the *Elaeis guineensis* species, accounting for 75.3% of the total markers. This mapping endeavor included a total of 473 million base pairs (Mbps) across the genome, offering valuable insights into the actual configuration of markers. Notably, the average distance between two consecutive markers in the *Elaeis guineensis* species is approximately 3.4 megabase pairs (Mbps). This information significantly contributes to our comprehension of the genetic architecture of this species [55].

Xiao et al. [55] conducted a comparative transcriptome analysis under cold stress conditions, which revealed the significant up-regulation of numerous genes, including a putative ICE1 ortholog, five CBF orthologs, 19 NAC transcription factors, and four cold-induced orthologs, all of which exhibited at least a two-fold increase in expression. Notably, the 5′ untranslated regions of Unigene21287 (a potential ICE1 ortholog) and CL2628.Contig1 (a NAC transcription factor) harbor SSR markers. By employing EST-SSR markers in *Elaeis guineensis,* researchers can enhance gene mapping accuracy. These markers, owing to their high polymorphism, facilitate the precise localization of genetic loci. The response of EST-SSR loci to low temperatures suggests their promising role in identifying trait-associated markers in oil palm. By exploring how these loci are activated under cold conditions, researchers may uncover genetic markers linked to crucial agronomic traits. This, in turn, could enable the development of more resilient and high-yielding oil palm varieties capable of thriving in diverse environmental conditions.

## 5. Marker-Trait Association (MTA) for Cold Tolerance

Marker-trait association (MTA) is a pivotal approach for uncovering genetic markers that correlate with specific plant traits. This methodology entails a thorough analysis of genetic variation within a population, aiming to pinpoint markers that consistently align with the trait of interest. In the context of oil palm, MTA holds immense potential for enhancing cold tolerance by identifying markers associated with cold-responsive genes. These markers can be leveraged for gene mapping, population structure analysis and, potentially, the identification of trait-associated markers specifically for cold tolerance in oil palm [55]. By employing MTA, breeders can selectively cultivate oil palm germplasm-carrying markers indicative of cold tolerance, thereby fostering the development of resilient varieties that exhibit enhanced oil productivity [56]. The MTA studies in oil palm have developed into the realm of genetic markers that are potentially linked to cold tolerance. For instance, Xiao et al. [55] discovered a significant number of 5791 gene-based Simple Sequence Repeats (SSRs) within the oil palm genome. Marker-assisted selection (MAS) techniques are invaluable tools that can greatly enhance cold tolerance in oil palm varieties. By harnessing these Molecular Technologies of Analysis (MTAs), breeders can precisely identify genetic markers that are associated with cold-tolerance traits. This targeted approach significantly reduces the need for lengthy and costly field trials, thus accelerating the breeding process. The implementation of MAS strategies not only speeds up the breeding cycle, but also ensures the development of oil palm varieties that are more resilient to cold stress and can thrive in challenging environments. However, it is worth noting that MTA studies for cold tolerance in oil palm are still in their early stages, necessitating further research to validate and refine the identified MTAs. The utilization of MAS for improving cold tolerance in oil palm remains an area that requires rigorous investigation to assess its effectiveness and efficiency in breeding programs. Since this approach is not yet widely used, we urgently need more studies to explore its potential benefits in enhancing cold-tolerance traits in oil palm varieties. Expanded research efforts in this field could provide crucial insights into the application of MAS for improving cold tolerance in oil palm breeding programs, ultimately leading to the development of more resilient and high-yielding varieties.

## 6. Genomics in Oil Palm Response to Abiotic Stress

Cao et al. [57] found that low temperatures have a detrimental impact on the growth and development of oil palm seedlings. The result observed an increase in malondialdehyde (MDA), a marker for membrane damage, and proline concentration in the leaves when seedlings were exposed to cold temperatures. These findings indicate that oil palm possesses certain response mechanisms to mitigate the adverse effects of low temperatures, such as elevated production of MDA and proline to prevent against membrane damage and cellular dehydration, respectively. Notably, cold-tolerant oil palm cultivars have been successfully cultivated in Kachin state, Myanmar, which is situated at a high elevation of approximately 25° N. These cultivars have demonstrated resilience to the region’s low temperatures, maintaining their vitality and producing healthy offspring. This accomplishment represents a significant milestone for the oil palm industry, broadening the crop’s potential growing regions and offering farmers a more diverse selection of cultivars. Furthermore, this achievement underscores the importance of selecting cultivars tailored to specific growing conditions to optimize yield and overall crop health. For instance, Agricultural Services Development Costa Rica could potentially develop cold-tolerant tenera hybrids by crossing DAMI deli with Cameroon and Tanzanian selections. These hybrids, with their enhanced cold tolerance, could be effectively cultivated at altitudes up to 1500 m in Ethiopia, Malawi, Kenya, Zambia, and Cameroon. This approach holds promise for improving the resilience of oil palm cultivation in these regions, enabling smallholder farmers to enhance their productivity despite cold temperatures. According to Yeap et al. [58], the gene EgRBP42, encoding a member of the plant heterogeneous nuclear Ribonucleoprotein family, exhibits a significant accumulation in oil palm leaves in response to various forms of abiotic stress, including salinity, drought, and cold. This observation indicates the potential beneficial role of this RNA binding protein in enhancing oil palm’s resilience under such stressful conditions. Furthermore, the study revealed that transgenic Arabidopsis plants overexpressing EgRBP42 exhibited early flowering and exhibited remarkable tolerance to a wide range of stresses, including heat, cold, drought, flood, and salinity. Moreover, these plants displayed improved post-stress recovery responses. Notably, the protein interacts with transcripts associated with diverse cellular processes, such as nucleocytoplasmic RNA transport, thereby implying its pivotal function in post-transcriptional regulation of stress response mechanisms. This finding underscores the potential of EgRBP42 in enhanced stress tolerance in oil palm and other crops.

## 7. Applications of Transcriptomics for Plant Sciences

Transcriptomics, the discipline devoted to studying the comprehensive set of RNA transcripts within a cell, tissue, or organism at a designated time point [59], holds significant potential for elucidating molecular mechanisms underlying stress responses, thereby fostering the creation of novel cultivars [60]. In the realm of plant sciences, transcriptomics promises numerous future applications. RNA-Seq, a high-throughput technique, has been instrumental in gene prediction, functional analysis, and the comprehension of gene ontology, all of which are essential for refining breeding methodologies [61]. Transcriptome data sheds light on alterations in gene expression during plant stress responses, underscoring the pivotal role played by both coding and noncoding RNAs in orchestrating defense mechanisms [62]. RNA-Seq has been instrumental in identifying functional genes, regulatory processes, and molecular mechanisms in non-model plants, thereby advancing breeding and cultivation practices. Furthermore, the integration of RNA-Seq data with other omic datasets has propelled plant regulomics, offering profound insights into genome-wide transcriptional regulation. More recently, advancements in single-cell RNA sequencing (scRNA-seq) have revolutionized our understanding of plant growth and development at the cellular level [63], heralding a new era of precision plant biology. Single-cell transcriptome technologies, including single-cell RNA sequencing (scRNA-seq) and single-nucleus RNA sequencing (snRNA-seq), offer profound insights into the distinct characteristics of each cell type, enabling the classification of cell types, tracing cell development trajectories, and deciphering gene regulatory networks in plants [64]. These studies are paramount to understanding the molecular mechanisms that govern plant development and their responses to environmental cues. Researchers often select specific technologies based on the tissue type required, whether it be cells or nuclei. Spatial transcriptomics, an emerging and promising technology, provides invaluable insights into the spatial arrangement of gene expression within tissues. It allows researchers to study gene regulatory networks in a holistic and spatially resolved manner. By preserving spatial information while profiling the transcriptome, spatial transcriptomics enables the identification of spatially coordinated gene expression patterns and an understanding of how these patterns contribute to the development, functionality, and response of plant tissues to various stimuli, including stress. This technology has the potential to revolutionize plant science research, enhancing our comprehension of plant development, stress responses, and adaptation mechanisms. By integrating spatial transcriptomics with other omics approaches, researchers can gain a more comprehensive view of plant biology and expedite the development of breeding strategies aimed at improving stress tolerance and overall plant development.

Transcriptomic analyses of oil palm under low-temperature stress have yielded profound insights into the genetic mechanisms that govern cold tolerance. Lei et al. (2014) [7] delved into the transcriptomic alterations in oil palm triggered by cold stress, revealing that the C-repeat binding factors (CBFs), belonging to the AP2/ERE family, might be pivotal in oil palm’s cold tolerance. Chen et al. [65] identified EgFAD8 genes in oil palm and demonstrated that the regulation of plastidial ω-3 fatty acid desaturases plays a crucial role in plant stress response. For instance, they found that low temperature up-regulates a FAD gene in Arabidopsis thaliana, highlighting the importance of FAD8 genes in responding to environmental stresses.

Beyond these comprehensive studies, research has zeroed in on specific genes and pathways integral to stress response. Notably, the transcription levels of CBF (C-repeat binding factors) are significantly increased by ICE (inducers of CBF expression), a type of MYC-type alkaline helix-loop-helix family transcription factor. Subsequently, CBF activates the expression of downstream Cold Responsive genes (CORs) by binding to specific cis-elements in the promoter region [66]. This pathway, which is conserved across many plant species, plays a vital role in cold acclimation. Moreover, the expression of CBF1/CBF3 genes in oil palm leaves was positively correlated with the expression of cold-response genes under cold stress conditions [11] (Figure 3), further highlighting the importance of this pathway in oil palm’s cold tolerance. In oil palm, transcriptomic studies have uncovered several pivotal gene families that are instrumental in the plant’s response to abiotic stresses, particularly low temperatures. Among these, the MYB gene family stands out as a crucial group of transcription factors involved in diverse biological processes, including stress response. In Arabidopsis, MYB15 has been identified as a key regulator that modulates the expression of CBFs (C-repeat binding factors), the master regulators of cold-responsive genes, under low-temperature stress [67]. Similarly, in oil palm, the EgMYB gene has emerged as a significant player in cold tolerance. Studies have shown that the overexpression of specific EgMYB genes, like EgMYB111 and EgMYB157, positively regulates the plant’s ability to tolerate abiotic stresses, including cold stress [68]. Zhou et al. [69] conducted a genome-wide screening in oil palm and identified 159 MYB genes, of which 20 were significantly up-regulated under low-temperature stress conditions. This finding suggests that EgMYB genes hold the potential to be targeted for genetic enhancement to improve cold tolerance in oil palm. In addition to the MYB transcription factors (TFs), the WRKY family of TFs has garnered significant attention for their involvement in cold resistance and hardiness in oil palm. Notably, WRKY1 and WRKY7 have been identified as regulators of these traits in the XJS30 and SJ64 cultivars under cold treatment [70]. Given their pivotal roles in various biological processes, including plant development and environmental stress resistance, the WRKY TFs represent crucial targets for studying and enhancing stress tolerance in oil palm. Many studies have been conducted to elucidate the role of different gene families in oil palm (Table 1).

Transcriptomic analyses in oil palm have offered profound insights into the molecular responses to cold stress. For instance, Xiao et al. conducted a study that revealed a set of expressed sequence tags (ESTs) exhibiting differential expression patterns in response to cold stress. Among these ESTs, 916 originated from expressed sequences that were up- or down-regulated by at least two-fold under cold stress conditions. These differentially expressed genes are likely involved in diverse biological processes, including signal transduction, transcriptional regulation, and metabolism, essential for the plant’s ability to respond and adapt to cold stress, ultimately influencing its survival and resilience. Moreover, the study underscores the potential applications of EST-simple sequence repeat (SSR) loci that exhibit inducible expression in response to low temperatures in identifying trait-associated markers in oil palm. This underscores the dual significance of transcriptomic studies: not only do they aid in understanding the molecular mechanisms underlying cold stress response, but they also facilitate the identification of genetic markers that can be leveraged in breeding programs to enhance cold tolerance in oil palm.

## 8. Application of Proteomics for Plant Sciences

Proteomics has profoundly enhanced our understanding of plant stress response pathways, unveiling the intricate molecular mechanisms plants employ to adapt to diverse environmental stresses. Researchers have harnessed mass spectrometry-based proteomics to identify stress-responsive proteins in plants exposed to drought, heat, and other stressors [74,75]. Advanced quantitative proteomics methods, such as data-dependent acquisition (DDA) and data-independent acquisition (DIA), have deepened our knowledge of protein abundance and post-translational modifications (PTMs) [76]. These techniques empower researchers to quantify protein levels and track PTM changes, offering critical insights into cellular processes and disease mechanisms. The enrichment of PTMs, achieved through techniques like metal oxide affinity chromatography (MOAC) and immobilized metal affinity chromatography (IMAC) for phosphorylation, has significantly broadened the scope of phosphoproteome coverage. This advancement aids in elucidating the regulatory mechanisms of various environmental signals in plants. By deciphering protein interactions and PTMs, proteomics enables the development of stress-tolerant crop varieties through targeted genetic modifications [77]. These comprehensive analyses have underscored the pivotal role of proteins in stress acclimation, revealing both common and unique protein response pathways across diverse plant species under varying stress conditions [78]. Additionally, proteomic techniques have been instrumental in identifying key biomarker proteins and PTMs that influence plant stress tolerance, paving the way for novel breeding strategies aimed at enhancing crop resilience under adverse conditions [75]. The integration of proteomics in crop breeding not only boosts nutritional value and yield but also plays a crucial role in addressing global food security challenges posed by unpredictable weather patterns and climate change.

Organelle proteomics analysis has recently emerged as a pivotal application of proteomics in plant sciences, offering insights into protein localization, organelle composition, dynamics, and functionality. This technique involves the purification of target organelles and the subsequent identification and quantification of proteins in enriched subcellular fractions [79]. Proteomics significantly contributed to our comprehension of the molecular mechanisms that underlie plant development and growth. By leveraging proteomic analysis techniques, researchers have been able to explore changes in protein expression across various stages of plant development, thereby identifying key proteins that are integral to crucial developmental events. Furthermore, proteomics has been instrumental in identifying enzymes and proteins involved in primary and secondary metabolism, encompassing carbohydrate, amino acid, and lipid metabolism, as well as the biosynthesis of specialized compounds like phytochemicals, pigments, and defense compounds. In summary, proteomics has provided invaluable insights into the molecular basis of plant stress responses, laying the foundation for targeted approaches to crop improvement.

Proteomics offers profound insights into how plant protein metabolism transforms under low-temperature stress and how these changes relate to cold tolerance. It is particularly valuable in identifying proteins synthesized under such stress, as they constitute essential elements for plants’ short-term adaptation. In the case of oil palm, proteomic responses to cold stress involve significant changes in protein abundance and functionality [11,80]. Cold stress elicits a complex response in plants, enhancing the expression of stress-responsive proteins while suppressing those linked to photosynthesis. This strategic adjustment enables plants to prioritize survival in harsh conditions and potentially foster cold resistance over time. By modifying protein expression patterns in response to cold stress, plants demonstrate their resilience and enhance their chances of survival in challenging environments [81]. Notably, the expression of key genes like CBFs correlates closely with the accumulation of osmolytes, such as proline and sucrose, suggesting their pivotal role in cold adaptation [82]. Proteomic analysis further reveals alterations in proteins involved in antioxidative defense mechanisms, stress responses, photosynthesis, and respiration [78]. Importantly, this study underscores the need to comprehend the molecular mechanisms underlying cold stress responses in oil palm to facilitate the development of cold-tolerant cultivars. The integrated omics approach offers profound insights into the molecular underpinnings of oil palm’s adaptation to low-temperature stress, providing crucial information for enhancing cold tolerance in this economically significant tropical crop. Utilizing both transcriptomic and iTRAQ-based proteomic techniques, in a previous study we investigated the response of three oil palm varieties, B × E, O × G, and T × E, to low-temperature stress (8 °C for 5 days). Our study revealed a differential expression pattern of genes (DEGs) and proteins (DEPs) in each variety. Specifically, the iTRAQ-based proteomic analysis showed significant changes in protein abundance, with 349 up-regulated and 657 down-regulated DEPs in B × E, 372 up-regulated and 264 down-regulated DEPs in O × G, and 500 up-regulated and 321 down-regulated DEPs in T × E, compared to control samples treated at 28 °C and 8 °C, respectively. Furthermore, our proteomic analysis identified proteins involved in stress, photosynthesis, and respiration in the three oil palm varieties. Notably, in the T variety, the abundance of stress-responsive proteins increased, while the levels of photosynthesis-related proteins decreased in response to low-temperature stress. This shift in protein abundance suggests that the T variety may develop cold tolerance as an adaptive response to cold stress [80]. By harnessing the power of proteomics alongside genomics and other biotechnological tools, we can adopt a combinatorial approach that accelerates gene discovery and enhances stress tolerance mechanisms in plants. Ultimately, this approach will lead to more effective crop improvement programs, particularly for economically vital crops like oil palm.

## 9. Application of Metabolomics for Plant Science

Metabolomics, a branch of “omics” research, delves into the intricate study of small molecules, named metabolites, within biological systems. In the realm of plant science, metabolomics serves as a powerful tool to decipher the metabolic alterations plants undergo in response to environmental stressors, encompassing drought, salinity, and temperature fluctuations [83]. By precisely identifying and quantifying metabolites in plant tissues, researchers can glean profound insights into the biochemical pathways that are perturbed by stress and the molecular mechanisms underlying plants’ adaptive responses. Metabolomics relies on a diverse array of analytical techniques, including mass spectrometry and nuclear magnetic resonance spectroscopy, to detect and quantify metabolites. The resulting data are then subject to rigorous statistical and bioinformatics analysis, aiming to uncover patterns and correlations between metabolites and environmental variables. However, one of the inherent challenges in metabolomics lies in the vast array of metabolites present in plant tissues, numbering from hundreds to thousands. To tackle this challenge, researchers often employ targeted or untargeted approaches, allowing them to analyze specific metabolite sets or conduct a comprehensive evaluation of all metabolites in a given sample, respectively.

Metabolomics has indeed emerged as a formidable tool for investigating the impact of abiotic stresses, particularly drought, on oil palm. A study conducted by Neto et al. [84] aptly exemplifies the prowess of metabolomics in analyzing the metabolic alterations of oil palm under drought stress. Their findings revealed a gradual metabolic shift from the 7th to the 14th day of stress exposure, suggesting a time-dependent adaptive response to the imposed stress. Notably, the study pinpointed several pivotal metabolic pathways that underwent significant alterations due to drought stress. These include starch and sucrose metabolism, glyoxylate and dicarboxylate metabolism, alanine, aspartate, and glutamate metabolism, arginine and proline metabolism, as well as glycine, serine, and threonine metabolism. These pathways are well-recognized for their crucial roles in stress tolerance and osmotic regulation in plants, further underscoring their significance in oil palm’s response to drought stress. The metabolomic response of oil palm to cold stress remains an under-explored area, yet insights gained from research on other plant species offer promising leads into potential metabolic alterations. In species such as winter-hardy wheat, for instance, cold stress triggers the accumulation of specific metabolites that safeguard plant cells from cold-induced damage and preserve membrane stability.

Zhao et al.’s [82] research into the metabolic changes in a winter-hardy wheat cultivar exposed to cold stress revealed that 745 genes were up-regulated after freezing treatment, indicating a substantial alteration in gene expression during cold acclimation and freezing. These alterations in gene expression likely influence the accumulation of specific metabolites, thereby enabling the plant to withstand cold stress. The effects of cold acclimation and freezing treatments on plant metabolite profiles are noteworthy, particularly in pathways such as the ABA/JA phytohormone signaling pathway and proline biosynthesis. Low-temperature stress triggers significant changes in metabolites, with crucial pathways involving ABA/JA signaling and proline biosynthesis playing pivotal roles in bolstering cold tolerance in plants like wheat. The ABA and JA signaling pathways collaborate to activate stress-responsive genes, enabling plants to effectively respond to cold stress by regulating essential physiological processes. Meanwhile, proline biosynthesis plays a crucial role in sustaining cellular function under cold conditions [85]. Similarly, metabolome analyses were conducted to explore the response of pumpkin (*Cucurbita maxima*) to cold stress [86]. This study identified 114 differentially expressed metabolites, focusing primarily on carboxylic acids and derivatives as well as organ oxygen compounds. The revelation of a series of potential metabolites and their corresponding genes underscores a comprehensive regulatory mechanism. These findings offer novel insights into the molecular mechanisms governing the response to cold stress in *C. maxima* [86]. Although research on oil palm’s response to cold stress is scarce, it is plausible that similar metabolic alterations occur in this species. Further research is imperative to comprehensively understand the molecular mechanisms underlying oil palm’s adaptation to low-temperature stress and to devise strategies that can enhance its stress tolerance and productivity in cooler climates.

## 10. Conclusions

In recent years, multi-omics approaches have gained significant momentum in elucidating the complexities of low-temperature stress in oil palms. While these approaches have advanced our understanding of the molecular mechanisms underlying oil palm’s response to cold stress, several challenges and limitations persist. Firstly, the absence of a reference genome for oil palm hinders the precision and efficiency of transcriptomic and proteomic data analysis. Secondly, the intricate nature and variability of oil palm metabolites pose challenges in identifying and quantifying all metabolites involved in the low-temperature stress response. Thirdly, the integration and interpretation of multi-omics data demand sophisticated bioinformatics tools and expertise, which are not widely accessible to researchers in this field.

New advances in high-throughput techniques have given rise to the profiling of different omics data levels in biological samples, i.e., genomics (DNA sequence level), transcriptomics (RNA expression level), proteomics (protein level), and metabolomics (metabolite level). These multiple levels of biological data provide the opportunity to integrate them into multi-omics data to give a more holistic and comprehensive perspective of the biological systems. The future outlook for oil palm omics research is promising, with a focus on addressing the challenges posed by climate change, disease resistance, and environmental sustainability. Future research must prioritize the development of a comprehensive reference genome for oil palm. This will significantly enhance the accuracy and efficacy of both transcriptomic and proteomic data analysis. Moreover, advancements in metabolomics technologies, such as non-targeted metabolomics and metabolite imaging, will enable the identification and quantification of a broader spectrum of metabolites involved in the response to environmental conditions. The integration of multi-omics data, utilizing techniques such as machine learning and network analysis, will offer a more holistic and systems-level understanding of the molecular mechanisms in oil palm.

In summary, multi-omics approaches have shed valuable light on the molecular mechanisms that govern oil palm’s responses to low-temperature stress. However, further research is necessary to integrate multi-omics data and validate potential candidate genes and proteins, ultimately leading to the development of strategies to enhance oil palm’s tolerance to low-temperature stress.

## Figures and Tables

**Figure 1 ijms-25-07695-f001:**
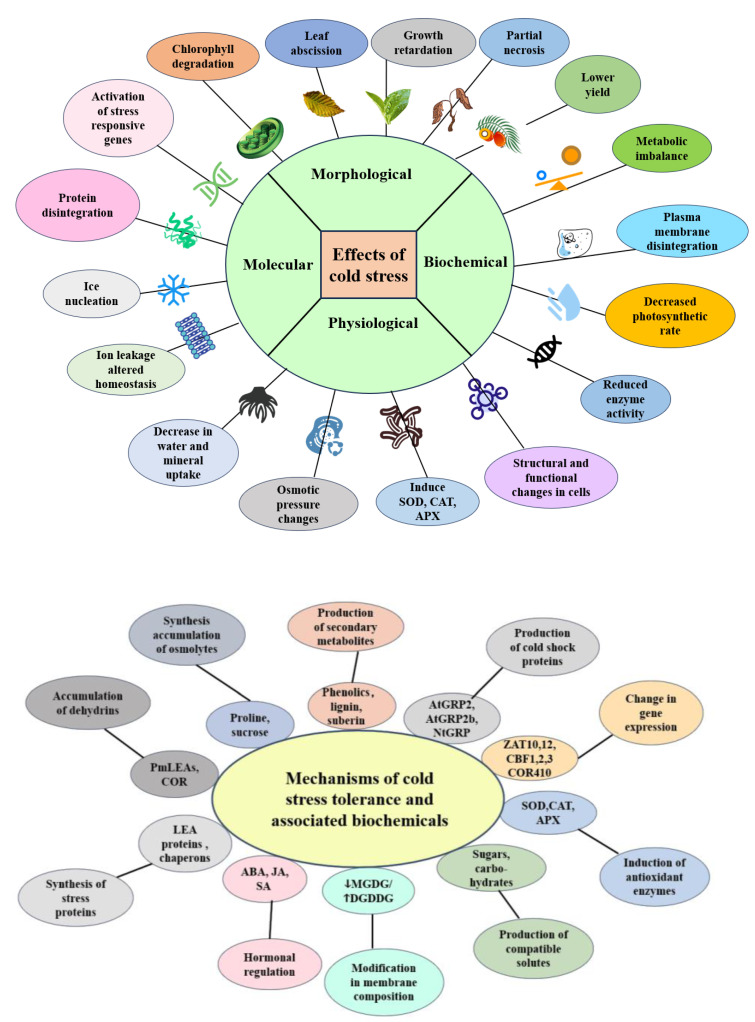
Effects of cold stress and tolerance mechanisms on oil palm: different physiological, morphological, biochemical, and molecular changes take place during stress conditions. Arrow: Cold acclimation-induced membrane changes are manifested through shifts in the concentration of membrane lipids. One change is that the concentration of monogalactosyldiacylglycerol (MGDG) tends to decrease in tandem with an increase in digalactosyldiacylglycerol.

**Figure 2 ijms-25-07695-f002:**
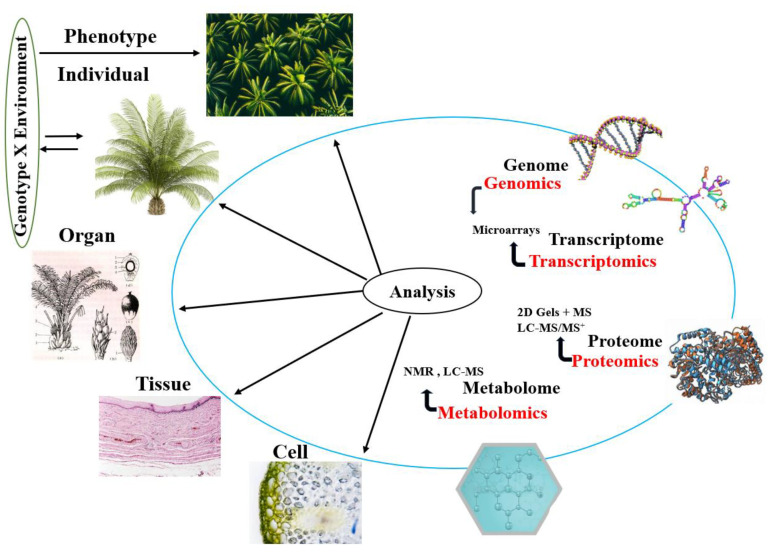
Omics approaches for oil palm improvement: techniques help unravel the complexities of agriculturally important traits.

**Figure 3 ijms-25-07695-f003:**
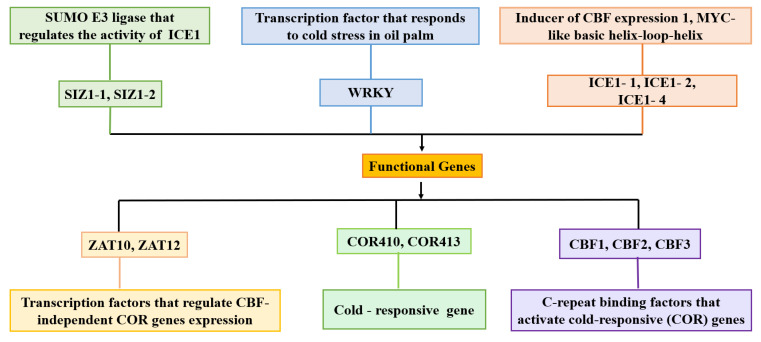
Functional genes that regulate cold temperature stress in oil palm.

**Table 1 ijms-25-07695-t001:** Overview of Abiotic Stress-Related Genes in Oil Palm.

Gene Family	Gene Name	Temperature	Descriptions	Reference
CBF	*CBF1*, *CBF2*	4 °C	At 4 °C, CBF1 (R^2^ = 0.68), CBF2 (R^2^ = 0.680.91) gene expression was significantly linked to sucrose.	[11]
	*CBF1*, *CBF3*	12 °C	Expression of CBFs were clear link with ICE1, SIZ1, ZAT10, COR413, and ZAT12 expression.	
MYB	*EgMYB111*, *EgMYB157*	8 °C	Overexpression of EgMYB111 and/or EgMYB157 significantly increases abiotic tolerance in transgenic Arabidopsis plants.	[68]
	*EgMYB38*, *EgMYB43*, *EgMYB57*, *EgMYB76*, *EgMYB82*, *EgMYB91*, *EgMYB104*, *EgMYB106*, *EgMYB111*, *EgMYB127*, *EgMYB133*, *EgMYB146*, *EgMYB151*, *EgMYB155*		A total of 14 MYB genes were significantly up-regulated under all abiotic stress conditions (cold, salinity, and drought); EgMYB146, EgMYB151, and EgMYB155 were all significantly up-regulated	[69]
WRKY	*EgWRKY03*, *EgWRKY06*, *EgWRKY07*, *EgWRKY11*, *EgWRKY16*, *EgWRKY25*, *EgWRKY26*, *EgWRKY28*, *EgWRKY29*, *EgWRKY35*, *EgWRKY52*, *EgWRKY59*, *EgWRKY61*, *EgWRKY72*, *EgWRKY76*, *EgWRKY80, EgWRKY88*	8 °C	17 EgWRKYs with greater than two-fold change in expression under cold stress	[70]
AP2/ERF/RAV	*EgAP2.15*, *EgAP2.34*, *EgERF23*, *EgERF104*, *EgERF130*	8 °C	Increase expression of AP2/ERF genes in response to cold exposure.	[71]
bZIP	*EgbZIP1*, *EgbZIP4*, *EgbZIP27*, *EgbZIP44*, *EgbZIP52*, *EgbZIP68*, *EgbZIP77*, *EgbZIP85*, *EgbZIP86*, *EgbZIP89*, *EgbZIP95*	NA	The bZIP genes were up-regulated in response to cold, salt, or drought stress, suggesting that EgbZIP plays a significant role in stress response.	[72]
ARF	*EgARF4*, *EgARF5*, *EgARF6*, *EgARF9*, *EgARF10*, *EgARF12*, *EgARF13*, *EgARF15*, *EgARF21*, *EgARF22* (up-regulated)	8 °C	Different types of abiotic stresses can induce the expression of EgARFs (cold, drought, and salt). The ARF gene functional investigations in oil palm serve as a genetic resource platform for oil palm abiotic stress resistance breeding.	[73]
	*EgARF1*, *EgARF3*, *EgARF8*, *EgARF14*, *EgARF17*, *EgARF18*, *EgARF19*, and *EgARF20* (down-regulated)			

## Data Availability

The data presented in this study are available on request from the corresponding author.

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
