# Peer review of "Multi-Omics Approaches in Oil Palm Research: A Comprehensive Review of Metabolomics, Proteomics, and Transcriptomics Based on Low-Temperature Stress"

_ijms, 2024, doi:10.3390/ijms25147695_

Round 1

Reviewer 1 Report

Comments and Suggestions for Authors

The first point; this manuscript is Review (not article)

The review is well organized and English is fine, however some modifications are needed in the structure and references section (some references are missing). 

All the best

Comments on the Quality of English Language

English is fine

Reviewer 2 Report

Comments and Suggestions for Authors

In this review article authors have focused the recent advancements in multi-omics studies on oil palm under low temperature stress conditions. This review highlights changes in metabolite profiles, protein expression, and gene transcription, and the potential of integrating data to reveal novel insights. The review emphasizes the challenges and prospects of multi-omics approaches in oil palm research, providing a roadmap for future investigations. The reviewer appreciates the authors have done comprehensive review on advancements in multi-omics studies on oil palms under low temperature stress conditions. However, the reviewer has major suggestions regarding this review. Thus, the authors need to consider the following comments to improve the quality of this manuscript.

1.         Improve the review's objectives or main theme in the introduction's last paragraph.

2.         Please enhance all the figure resolutions because the details/information are unclear.

3. In Figure 1, Mechanisms of cold stress tolerance can be improved with more details like Changes in gene expression (Example genes) and production of compatible solutes (example).

4. The genomic section looks too general and shallow. Improve it with specific details.

5.         Improve the coherence and logical flow of the article by ensuring smooth transitions between paragraphs and sections. Consider restructuring the article to improve the connection between different section to a more appropriate location within the text. The Transcriptomics section should come after the genomics section. (Genomics, Transcriptomics, Proteomics and metabolomics)

6.         Line 249: Remove the closing bracket ‘)’

7.         The authors are asked to carefully check the formatting, punctuation, space errors, symbols, etc., in the entire manuscript.

8.         Abbreviate the QTL and others in the first mention. Example: Line 144 represented only QTL, and line 250: Quantitative Trait Loci (QTLs). Revise it.

9.         Gene names should be in italics format

10.     If possible, create one table with multi-omics information on oil palm.

11.     Many sections seem authors just collected and collated the information from the literature. Authors should write one or two lines about how this study will be helpful for the reviewed issue in current and future scenarios in the last sentence of the paragraph, where it is applicable.

Comments on the Quality of English Language

Please refer the author comments

Round 2

Reviewer 1 Report

Comments and Suggestions for Authors

The manuscript is greatly improved and comments are provieded. However minor corrections are still needed (see attached)

All the best

Comments on the Quality of English Language

Author Response

Comment: regarding heading or sub heading

Reply:Thank you for your comment regarding the headings and subheadings in my manuscript. We review the headings and subheadings to ensure they are clear, concise, and consistent throughout the manuscript.The changes we have made to our paper are clear in the revised version as tracked changes. We make the necessary adjustments to improve the organization and readability of the manuscript.

Reviewer 2 Report

Comments and Suggestions for Authors

The authors have suitably incorporated my comments in the revised manuscript. Now, the overall quality of the manuscript has significantly improved. Therefore, I recommend the manuscript be accepted for publication in its current form. 

Author Response

Dear Reviewer

We would like to express our sincere gratitude for your thorough and constructive review of the manuscript " Multi-Omics Approaches in oil palm Research: A Comprehensive Review of

Metabolomics, Proteomics, and Transcriptomics based on low temperature stress". Your comments and suggestions have been invaluable in improving the quality of the manuscript.

We are pleased to inform you that the authors have successfully incorporated your comments in the revised manuscript, and the overall quality of the manuscript has significantly improved. We appreciate the time and effort you have invested in reviewing the manuscript, and we are grateful for your recommendation to accept the manuscript for publication.

Thank you again for your expertise and support. We look forward to continuing to work with you in the future.

Best regards,

Jerome Jeyakumar